# The Role of Alternative Splicing Factors, DDB2-Related Ageing and DNA Damage Repair in the Progression and Prognosis of Stomach Adenocarcinoma Patients

**DOI:** 10.3390/genes14010039

**Published:** 2022-12-23

**Authors:** Xinshu Wang, Zhiyuan Huang, Lei Li, Yuntong Yang, Jiyuan Zhang, Li Wang, Jian Yuan, Yunhui Li

**Affiliations:** 1Postgraduate Training Base of Jinzhou Medical University, Shanghai East Hospital, Shanghai 200120, China; 2Research Center for Translational Medicine, East Hospital, Tongji University, School of Medicine, Shanghai 200120, China; 3Department of Biochemistry and Molecular Biology, Tongji University School of Medicine, Shanghai 200120, China; 4Ji’an Hospital, Shanghai East Hospital, Ji’an 343000, China

**Keywords:** ageing, alternative splicing, DNA damage response, STAD, prognosis, DDB2

## Abstract

DNA damage response is a key signal transduction pathway in triggering ageing and tumor progression. Abnormal alternative splicing (AS) is associated with tumors and ageing. However, the role of AS factors associated with DNA damage repair and ageing in stomach adenocarcinoma (STAD) remains unclear. We downloaded the percentage of splicing (PSI) values for AS in STAD from the TCGA SpliceSeq database. The PSI values of DNA repair gene AS events were integrated with STAD patient survival data for Cox regression analysis. The prediction model for the overall survival (OS) was constructed by the clinical traits. The tumor immune microenvironment was analyzed by CIBERSORT and ESTIMATE. We detected 824 AS events originating from 166 DNA repair genes. Cox regression analysis provided 21 prognostic AS events connected with OS statistically, and a prognostic prediction model was constructed. The expression of these AS factors was higher in STAD tumors. *DDB2* high senescence levels were associated with active immune responses and better survival in STAD patients. We built a novel prognostic model founded on DNA repair genes with AS events and identified that *DDB2* may be a potential biomarker to apply in clinics.

## 1. Introduction

The human genome DNA is exposed to hundreds and thousands of exogenous and endogenous threats every day. After DNA is damaged, mounting DNA damage response factors gather at the damage site to repair the damage. If the damage is not properly repaired, cells may lead to ageing, apoptosis, and tumorigenesis [1]. Tumors are considered as an ageing disease. As a process of tumor suppression, the ageing of tumor cells prevents tumor cells proliferation, inhibits tumor progression, and regulates antitumor therapies [1].

Alternative splicing (AS) is a common form of gene expression regulation and a posttranscriptional mechanism, and 92–95% of human exons have undergone AS [2]. Changes in the expression of splicing factors are found in 50% of age-associated alternative splicing [1]. Moreover, aberrant AS regulation events are extremely linked to tumorigenesis and development, including gene mutation at the position where splicing occurs, splicing factor mutation or abnormal expression, and systemic splicing abnormalities in tumor cells. The variation in alternative splicing can cause the non-expression of tumor suppressor genes or induce the production of specific oncogenic proteins [3,4]. The human tumor suppressor TP53 controls DNA damage-induced cellular senescence. Recent research indicated that TP53 was found as an alternative splicing of cellular senescence markers through the DNA damage response pathway [5,6]. Tumor-specific AS events, related to DNA damage and cellular senescence, may be developed as prognostic and diagnostic targets and provide personal therapeutic strategies.

STAD is a subtype of gastric cancer. High histological heterogeneity and a lack of early detection have led to poor prognosis and make the survival rate of patients low [7,8,9]. Previous studies have identified abnormal DNA repair pathways that affect the occurrence and progression of STAD. DNA repair polymorphisms contribute to genetic susceptibility and increase the risk of stomach adenocarcinoma [10,11]. There are no related reports centering on ageing and DNA repair-related alternative splicing in STAD. 

In the present study, we focused on the AS of DNA repair genes in a TCGA STAD cohort. We screened out prognostic AS events of DNA repair genes and constructed a prognostic prediction model. More importantly, we identified a novel AS factor DDB2, which may provide potential biomarkers for prognosis and tumor therapy in STAD.

## 2. Method

### 2.1. Acquisition of Alternative Event Data and Transcriptome Seq Data with Clinical Features

The percentage spliced-in (PSI) values of alternative splicing events of stomach adenocarcinoma were gained from the TCGA SpliceSeq database (https://bioinformatics.mdanderson.org/TCGASpliceSeq/, accessed on 12 May 2021), which collected alternative mRNA splicing from the TCGA cancer database [12]. Transcriptome sequencing data of stomach adenocarcinoma specimens retrieved from patients who suffered from stomach cancer with their clinical traits were obtained from the TCGA database. Seventy-five percent of samples with PSI values were set as splice event filters. DNA repair genes were retrieved from a widely acknowledged gene list [13] to screen the alternative splicing events, which were displayed by UpSet plot. Alternative splicing events were integrated with the overall survival data of STAD patients for further analysis by Perl software.

### 2.2. Construction and Substantiation of the Prognostic Model

The missing PSI values for alternative splicing events were filled by KNN algorithms, and univariate Cox regression was applied to sort the prognosis-significant DNA repair for alternative splicing events. UpSet plots and forest plots were applied to present these prognostic alternative splicing events with their hazard risk and a 95% coincidence interval. A volcano plot was applied to present z-score normalized coefficients and the *p* value of each prognostic DNA repair alternative splicing event presented by the Wald χ^2^ test in univariate Cox regression analysis. They were combined to construct a multivariate Cox regression model, which was optimized by Lasso regression to avoid overfitting. Prognostic DNA repair gene alternative splicing events are displayed by a forest plot. Risk scores were computed according to the coefficients of prognostic DNA repair alternative events retained in the multivariate Cox regression model and their PSI values ($\mathrm{riskscore}=h_{0}\left(t\right)\exp(\overset{n}{\underset{j=1}{\sum }}{\mathrm{Coef}}_{j}\times {\mathrm{PSI}}_{j})$, *n* = quantity of picked genes, Coef_*j*_ is the coefficient of every DNA repair gene alternative splicing event, PSI_*j*_ is the PSI value of every DNA repair gene alternative splicing event, and *h*_0_(t) is the bottom line risk function). Stomach adenocarcinoma patients were divided into a high-risk group and a low-risk group in terms of the median risk score. The two groups’ overall survival was compared by the log-rank test in Kaplan-Meier analysis. The prognostic prediction value of other clinical traits and the risk score were compared by receptor operator curves. The time-dependent ROC curves for risk score are depicted at 1 year, 3 years, and 5 years, and time-dependent ROC curves for the risk score and other clinical characteristics were drawn at 1 year by the time ROC package in R software. The area under the ROC curve represents the prediction feasibility of each predictor, which extends from 0.5 to 1. A larger AUC indicates better sensitivity and better specificity of the predictor. The risk score was combined with age, sex, tumor stage, tumor grade, T stage, M stage, and N stage to perform univariate Cox regression and multivariate Cox regression to prove whether the risk score is an independent prognostic indicator. Together, we established a convenient model for the prognostic prediction of STAD patients. The prognostic model for STAD patients was presented by a nomogram whose calibration and discrimination were validated by a calibration curve and C-index for clinical application. Bootstrapping validation with 1000 bootstrap resamples was applied to draw a calibration curve for this nomogram. The nearer the distance from the diagonal to the calibration curve, the higher the calibration accuracy.

### 2.3. Retrieval of the Differently Expressed Prognosis-Related Genes and Analysis of Immune Infiltration

The expression level profile of host genes of prognosis-associated alternative splicing events in stomach adenocarcinoma specimens and corresponding stomach tissues was retrieved from the TCGA database. The expression profile was FPKM (fragments per kilobase of exon per million fragments mapped reads) normalized transcriptome RNA-seq data. The Wilcoxon rank-sum test was applied to retrieve differentially expressed DNA repair genes. A *p* value < 0.05 was set as the cut-off point to sort DNA repair genes for immune infiltration analysis.

### 2.4. Analysis of Tumor Mutation Burden (TMB)

Simple nucleotide variation data of STAD were downloaded from TCGA. They were integrated into one maf file by Perl software. The “Maftools” package in R software 4.1.0 was applied to display the mutation distribution. TMB scores between high expression of *DDB2* and low expression of *DDB2* were presented by box plot.

### 2.5. Analysis of the Features of Immune Microenvironment

The degree of tumor purity and the immunological infiltration level were evaluated by the ESTIMATE algorithm (https://bioinformatics.mdanderson.org/estimate/, accessed on 2 December 2022), which regarded gene expression profiles as signatures for stromal and immune score estimation [14]. The ESTIMATE score is the total of the stromal score and immune score, which revealed tumor purity and the immunological infiltration level. The FPKM normalized RNA-seq expression profile of STAD specimens was gained from the TCGA database to perform ESTIMATE calculation. STAD patients were separated into a high-expression-level group and a low-expression-level group by genes to compare tumor purity and immune infiltration.

Considering the density relationship with DNA repair and immune pathway functions in the cancer microenvironment, CIBERSORT was applied to evaluate the immune landscape. The procedure for using the CIBERSORT (http://cibersort.stanford.edu/, accessed on 2 December 2022) package in R software 4.1.0 followed that of Newman et al. [15] according to deconvolution, which has been proven to be able to quantify the gathering of immune cells in many cases. 

### 2.6. Analysis of the Correlation between the PSI Value of Prognostic Alternative Splicing Events and Clinical Features

The risk score and PSI value of each prognosis-related alternative splicing event between the low-risk group and the high-risk group were compared by *t* test or Kruskal-Wallis test depending on the quantity of categories of clinical features. The PSI value and risk score of STAD samples are presented by box plots with colored dots.

### 2.7. IHC Staining and Analysis

IHC samples were purchased from SHANGHAI OUTDO BIOTECH Co., Ltd., ShangHai, China. which included 30 STAD tissue samples. We used NELT2(Gene Tex GTX132565) PRKDC (ABclonal A7716) and RAD1(ABclonal A6841) antibodies for IHC staining. The tumor sections were deparaffinized, rehydrated, and placed in 0.3% hydrogen peroxide methanol for 20 min, then blocked with goat serum and incubated overnight with antibodies, rinsed, incubated with biotinylated rabbit polyclonal secondary antibody, stained with diaminobenzidine solution to visualize the HRP, and mounted. For quantification of the IHC results, the proportion of positives and negatives was analyzed. 

## 3. Results

### 3.1. Landscape of AS Event Profiles in the STAD Cohort

An analysis flowchart is shown in Figure 1a. The expression profile obtained from the TCGA database consisted of 407 samples in total, which included 375 stomach adenocarcinoma specimens and 32 corresponding paracarcinoma stomach tissues. Clinical information downloaded from TCGA database included STAD patients’ clinical features, which were listed in Table 1. The percent spliced-in (PSI) values expression profile of alternative splicing events was downloaded from the TCGA SpliceSeq database, which included 415 stomach adenocarcinoma specimens and 37 corresponding paracarcinoma stomach tissues. AS events were separated into seven splicing modes, including alternate promoter (AP), alternate acceptor site (AA), alternate terminator (AT), alternate donor site (AD), exon skipping (ES), mutually exclusive exons (ME), and retained intron (RI) (Figure 1b). A total of 219 DNA repair genes were retrieved from a widely acknowledged gene list [13] to screen the alternative splicing events, and 166 DNA repair genes were equipped with recorded an alternative splicing event in the PSI value profile. A total of 824 AS events originating from 166 genes were included in this study, and the top 3 splicing types accounted for approximately 72.0% of the total. They were ES, AP, and AT, with proportions of 43.6%, 16.1%, and 12.3%, respectively (Figure 1c). However, for the possibility of various splicing modes for a single gene, we built UpSet plots to analyze interactive sets of seven kinds of AS events quantitatively. As shown in Figure 1d, an UpSet plot is presented for DNA repair-correlated alternative splicing events, and most genes had more than one AS event.

### 3.2. The Alternative Splicing Events of DNA Repair Genes were Correlated with the Overall Survival of STAD Patients

The PSI value of tumor tissues was integrated with STAD patient survival data for Cox regression analysis. After the match of patient’s survival data and the PSI profile of DNA repair genes’ alternative splicing events by patient’s TCGA ID number, 504 patients were involved in Cox regression analysis. Cox regression analysis provided 21 prognostic alternative splicing events whose *p* value was less than 0.05 in the Wald χ^2^ test. The prognosis-related alternative splicing events are presented as an UpSet plot in Figure 2a. As displayed by the UpSet plot diagram, one gene probably experiences two or three sorts of AS events that were importantly connected with the survival of patients. The forest plot of prognosis-related alternative splicing events is presented in Figure 2b. A sum of 21 AS events were discovered to be greatly connected with the overall survival of STAD patients, including 9 ES events, 4 AP events, 3 AD events, 3 AT events, 1 AA event, 1 RI event, and 0 ME events. Notably, one gene may have more than two survival-related splicing events in STAD patients. To expand the prognostic predictors for STAD patient survival, important survival-related AS events were selected as candidates (ERCC1|50447|AD, TREX1|64682|RI, RAD51B|28109|AT, UVSSA|68487|AP, PARPBP|24045|AA, MBD4|66720|AD, SETMAR|62997|ES, OGG1|63171|ES, SETMAR|63000|ES, RAD51D|40268|ES, RAD1|71734|AP, RAD1|71736|AP, NEIL2|82636|AD, NBN|84397|ES, PRKDC|83791|ES, NUDT1|78606|AP, GTF2H1|14605|ES, MLH1|63932|AT, MLH1|63930|AT, DDB2|15675|ES, and RAD17|72362|ES). Then, a subset of AS events that are overlapped among the seven AS sorts in STAD patients was studied further. The volcano plot of prognosis-related alternative splicing events is presented in Figure 2c, which shows the z-score corrected coefficients of prognosis-related alternative splicing events and their *p* values provided by the Wald χ^2^ test in univariate Cox regression.

We established a risk score signature by least absolute shrinkage and selection operator (LASSO) Cox analysis of 16 prognostic signatures related to AS events. The sectional likelihood deviance for the LASSO coefficient profiles of 16 prognostic signatures is presented in Figure 2d,e. Significant prognostic AS events were preserved in the multivariate Cox regression model, which consisted of ERCC1|50447|AD, TREX1|64682|RI, RAD51B|28109|AT, UVSSA|68487|AP, PARPBP|24045|AA, MBD4|66720|AD, SETMAR|62997|ES, OGG1|63171|ES, SETMAR|63000|ES, RAD51D|40268|ES, RAD1|71734|AP, NEIL2|82636|AD, PRKDC|83791|ES, MLH1|63932|AT, DDB2|15675|ES, and RAD17|72362|ES (Figure 2f). Thus, it may be valuable to study these DNA repair-related survival-related AS events, which might supply novel therapeutic targets for STAD patients.

### 3.3. DNA Repair-Related Alternative Splicing Predicted the Prognosis of STAD Patients

The risk score was computed in accordance with the PSI value and coefficients of these 16 prognostic alternative splicing events. The survival of high-risk patients and low-risk patients separated by the median risk score was analyzed by Kaplan-Meier analysis. As shown in Figure 3a, the signatures could efficiently distinguish the low-risk groups and the high-risk groups patients from total patients. The *p* value presented by the log-rank test was less than 0.001, which explained that the discrepancy of overall survival between the high-risk and low-risk groups was great statistically. The distribution of the PSI value of prognosis-associated DNA repair AS events between the high-risk group and the low-risk group is displayed in a heatmap. The PSI value of RAD51D|40268|ES was higher evidently in the low-risk group, which might indicate its protective function in the progression of STAD (Figure 3b). The risk score estimated by DNA repair correlated alternative splicing events was able to recognize STAD patients with a significantly high risk of death (Figure 3c), which displayed the risk scores of STAD patients ranking from low risk to high risk. The median was utilized as a demarcation standard to separate the high-risk and low-risk groups. The prognostic hazard scatter plot for the STAD patients showed the distribution of the risk scores with overall survival. The results reveal that patients with a low-risk score had a longer survival time and that the risk score signature had strong prognostic value (Figure 3d).

### 3.4. Validation of the Risk Score and the Nomogram Grounded on Important Prognosis-Related DNA Repair Alternative Splicing Event

Then, the effect of prognostic signatures such as other clinical traits and the risk score in prognosis prediction was assessed by time-dependent ROC (receiver operator characteristic) curves. The risk score (AUC = 0.736) had a significantly better survival prediction ability at one year than age (AUC = 0.582), gender (AUC = 0.516), grade (AUC = 0.554), and stage (AUC = 0.609) (Figure 4a). Furthermore, the risk score had a better prediction ability at 2 years (AUC = 0.770) than at 1 year (AUC = 0.736) and 3 years (AUC = 0.747) (Figure 4b).

The risk score and other clinical features (age, gender, grade, and stage) were combined to perform univariate Cox regression analysis and multivariate Cox regression analysis. In univariate Cox regression, risk score (HR = 1.500, *p* value < 0.001), stage (HR = 1.569, *p* value < 0.001), and age (HR = 1.025, *p* value = 0.004) were significantly related to overall survival (Figure 4c). In the multivariate Cox regression model, risk score (HR = 1.458, *p* value < 0.001), stage (HR = 1.568, *p* value < 0.001), and age (HR = 1.029, *p* value = 0.001) were recognized as independent prognostic predictors. The forest plot of univariate and multivariate Cox regression is presented in Figure 4d. Thus, the risk scores were related with the overall survival (OS) of STAD patients, and univariate or multivariate Cox regression analyses proved that risk score calculated by PSI value of DNA repair-related alternative splicing events may be considered as an independent prognostic indicator of STAD.

A nomogram was constructed to present the prediction model constructed with the risk score and other clinical traits, which was verified by the C-index and calibration curve. The C-index was 0.748, and the calibration curve is presented at 1 year, 3 years, and 5 years (Figure 4e,f). In summary, C-index and calibration curves signify that the nomogram could act as the predictor of the prognosis of STAD patients.

### 3.5. Clinical Correlation Analysis of Prognostic DNA Repair Genes

We applied PSI values to evaluate the relationship between TNM stage and the DNA repair genes’ alternative splicing events. Stomach adenocarcinoma with a higher clinical stage had a higher risk score (Figure 5a). STAD samples with lymphoid metastasis (higher N stage than N0 stage) had a higher risk score (Figure 5b). The PSI value of NEIL2-82636-AD was higher in advanced stage disease, when comparing stage I–II to stage III–IV disease (*p* value = 0.006) (Figure 5c). In T3–T4 stage patients, the PSI value of MLH1-63932-AT was higher than that in early T stage patients (*p* value = 0.018), implying its dangerous role in the development of STAD (Figure 5d). Furthermore, the PSI values of NEIL2-82636-AD and PARPBP-24045-AA were higher in M1 STAD tissues than in M0 stage STAD tissues (Figure 5e,f). Therefore, we draw a conclusion that the alternative splicing events of DNA damage repair genes might be a reason for the prognosis of patients with STAD.

### 3.6. Differentially Expressed Prognosis-Associated DNA Repair Genes

The expression levels of 11 host genes of prognosis-associated AS were retrieved to perform a distinction expression analysis between normal stomach tissues and STAD specimens, which is presented in Table 2.

There were 407 STAD patients in the HCC dataset of TCGA, including 375 STAD tissue samples and 32 adjacent normal tissue samples with mRNA sequencing data. The mRNA expression levels of the *RAD51D*, *PRKDC*, *PARPBP*, *RAD1*, *NEIL2*, *RAD51B*, *UVSSA*, *MBD4*, and *RAD17* genes in STAD tumor tissues were markedly higher than those in STAD tissues, and the difference was statistically significant (Figure 6a–f and Appendix A). To make further validation about the differentially expressed DNA repair genes in STAD, we applied immunohistochemical staining in stomach adenocarcinoma cancer tissue microarrays. Immunohistochemistry (IHC) was used to analyze the three genes (*PRKDC*, *NEIL2*, and *RAD1*) that had the most notable differences in STAD tissues compared with adjacent normal tissues (Figure 6g). *PRKDC*, *NEIL2*, and *RAD1* were expressed at a higher level in STAD tissues. The genes should be verified in larger-scale clinical research, and the function of these molecular biological genes deserves to be explored more deeply.

### 3.7. DDB2 Is the Potential Prognostic Biomarker in STAD Patients with Active Immune Responses and Better Survival

DNA damage repair response is a key signal transduction pathway in triggering ageing and tumors. Aiming to validate AS factors related DNA repair and ageing, we retrieved the intersection between the cell ageing gene list [16] and the host prognostic DNA repair genes in Table 1. *DDB2* was regarded as target gene, which is an essential mediator of premature senescence and DNA damage repair [17]. We analyzed the alternative splicing events of *DDB2* in STAD by the TCGA SpliceSeq dataset and found that the alternate splicing patterns of the *DDB2* gene are mainly ES (Figure 7a,b).

Through correlation analysis between the expression values of survival-related alternative splicing factors and the constructed prognostic models, higher mRNA levels of *DDB2* were observed in STAD patients compared with normal ones (including stages of STAD progression, nodal metastasis status, and gender) in the UALCAN dataset (*p* < 0.05) (Figure 7c and Appendix A). Additionally, patients with a high expression of *DDB2* showed significantly better prognosis (*p*-value < 0.05) (Figure 7d). As *DDB2* contributed to cell apoptosis and senescence, these results reveal that *DDB2* high senescence levels were associated with prolonged survival in STAD. 

Increased genomic instability and tumor mutation are common markers of ageing and cancer [18]. At first, we analyzed tumor mutation in STAD patients (Appendix A). Missense mutation is the most frequent mutation, and single-nucleus polymerase (SNP) took up the greatest ratio of variant types. We also assessed missense SNVs for overall mutant and C > T mutant accounts for more than 55% of all SNVs mutations. Tumor mutational burden (TMB) and microsatellite instability (MSI) are molecular characteristics in genomic instability, which are considered as markers for immunotherapy response. In tumor cells, MMR gene function loss (dMMR) indicates a defect in the ability to repair DNA replication errors, and more hypermutated tumors will occur. TMB and MSI will be increased, and relevant new antigens will be generated in tumor cells. Lymphocytes will be mobilized in order to inhibit tumor growth [19]. In Figure 7e–f, higher TMB and MSI scores were presented in the high levels of *DDB2*, which causes more malignant characteristics, which may explain the high levels of *DDB2* associated with prolonged survival in STAD.

To further investigate the impact of *DDB2* on tumor immunity, we applied the ESTIMATE algorithm to evaluate the ESTIMATE score, immune score, and tumor purity. We observed that the low *DDB2* expression level group had a low ESTIMATE score (*p* = 0.0021) and immune score (*p* < 0.0001) (Figure 7g,h). The high *DDB2* expression level group had a low tumor purity degree (*p* = 0.0021) (Figure 7i). To confirm the correlation between *DDB2* expression and its immune microenvironment, the CIBERSORT algorithm was utilized to analyze 22 kinds of immune cell profiles, especially the percentage of tumor-infiltrating immune subsets. The results indicate that the expressing level of *DDB2* was positively associated with the activated memory CD4 T cells, CD8 T cells, follicular helper T cells, and M1 macrophages (Figure 7j–m). The expression level of DDB2 is negatively correlated with the memory resting CD4 T cell and resting mast cells (Figure 7n,o). Then, the expression levels of immune checkpoint genes were compared between the high and low *DDB2* gene expression groups. CD80, ICOS, CD86, TNFRSF25, TNFSF9, and CD48 had much higher expressing levels in the high-expression group (Figure 7p). These results suggest that *DDB2* could be the potential prognostic biomarker in STAD patients with active immune responses and better survival.

## 4. Discussion

Human bodies undergo more than 50,000 DNA damages per cell every day, which are mainly caused by endogenous and external factors. Endogenous factors—such as DNA replication pressure, DNA recombination, reactive oxygen species or respiration and external factors, such as radiation or chemical mutagens—are a threat to DNA integrity [20]. Ageing is connected with the cumulative amassing of DDR accumulation and senescent cells [21]. Mechanical stress to tissues is also able to make genomes instable. Even if most lesions are rapidly eradicated, DNA damage will still accumulate and might lead to ageing more quickly [22,23]. Cellular senescence is related to ageing and cancer in vivo and has a confirmed tumor-suppressive function [24]. As a process of tumor suppression, cellular senescence prevents tumor cells proliferation, inhibits tumor progression, and regulates antitumor therapies. The occurrence of DNA damage and the engagement of the DNA damage response pathways are the common ground of cancer and ageing [25,26].

The DNA damage response system maintains genome homeostasis by initiating DNA repair and signaling. In particular, it is now increasingly evident that post-transcriptional regulation is involved in the DDR pathway [27,28]. Alternative splicing (AS) is a crucial process following transcription and allows cells to generate diverse isoforms of RNA and protein isoforms through the precursor messenger RNA (pre-mRNA) [29,30,31]. These different isoforms of mRNA can have diverse regulatory and functional properties [32,33]. In human cells, the majority of human genes undergo alternative splicing [34]. The abnormal splicing events contributed to cancer development and progression, such as survival, angiogenesis, proliferation, metastasis, and invasion of cancer cells. 

Abnormal AS has been identified as a driving factor for GC cancers [35,36,37]. For example, *EBNA1* plays a key role in misregulating cellular AS in EBV-negative and Epstein–Barr-virus-associated gastric cancer [35]. Shi et al. provided comprehensive landscape alterations to the global mRNA splicing signatures from stomach adenocarcinoma (STAD) and analyzed the association of the AS events and the survival of STAD patients [36]. In 2021, a report summarized seven types of aberrant splicing constructed prognostic signatures for GC patients [37]. More importantly, GO and KEGG pathway analyses identified the three most significant biological processes, including “DNA repair”. However, there is still a lack of global analysis of the regulatory mechanism between DNA damage repair and survival-related AS events in STAD patients.

Aiming to validate the AS-related DNA repair in STAD, we detected 824 AS events, totally originating from 166 DNA damage repair genes that were comprised in this research. Additional analysis provided 11 prognostic AS factors that were connected with the OS. In accordance with previous studies, the genes are associated with ageing and GC tumors. *DDB2* was a component of the damage-specific DNA-binding heterodimeric complex, which is involved in the occurrence and development of premature ageing and cancer by affecting nucleotide excision repair (NER) and cell apoptosis [38]. *DDB2* is an essential mediator of premature senescence [39]. Reducing the expression of *DDB2* can inhibit the Ras-mediated premature senescence response. MEF cells lacking *DDB2* expression were resistant to apoptosis and chemotherapeutic drugs induced by DNA damage [40,41]. Furthermore, *DDB2*-induced CDT2 degradation may be associated with apoptosis in cancer cells [17]. *DDB2* contributes to gastric carcinogenesis using high-throughput sequencing in patients with GC [42]. The *NBN* gene is the main member of the MRN complex (*MRE11/RAD50/NBS1*), which recruits from the DNA damage site in the early stages [43,44]. Nijmegen breakage syndrome caused by mutations and variants in the *NBN* gene lead to susceptibility to gastric cancer [45]. Moreover, NBN has been proven in conjunction with genes to likely be involved in human ageing, such as Sirt1, WRN [46]. *PRKDC*, a crucial gene in the NHEJ pathway, encodes the catalytic subunit (*DNA-PKcs*), which takes participation in DSB repair in order to maintain genome stability [47]. During ageing, metabolic and fitness decline may be due to *PRKDC* accumulation of DNA breaks [48,49]. In gastric cancer, PRKDC was identified as the top upregulated DNA damage repair gene, and high expression of *PRKDC* was associated with poor survival. These results are consistent with this study [50].

In this study, a new prognostic model built on DNA damage repair genes with AS events was constructed for STAD patients, proving that survival-associated AS features may serve as predictors of prognosis in STAD. This research indicates the key DNA repair factors played a significant role in STAD-associated AS events. Furthermore, we found a novel AS factor *DDB2* associated with ageing and DNA repair, which was related with high TMB, high MSI, active immune responses, and better survival. Additionally, *DDB2* maybe provide potential biomarkers for prognosis and tumor therapy in STAD.

## Figures and Tables

**Figure 1 genes-14-00039-f001:**
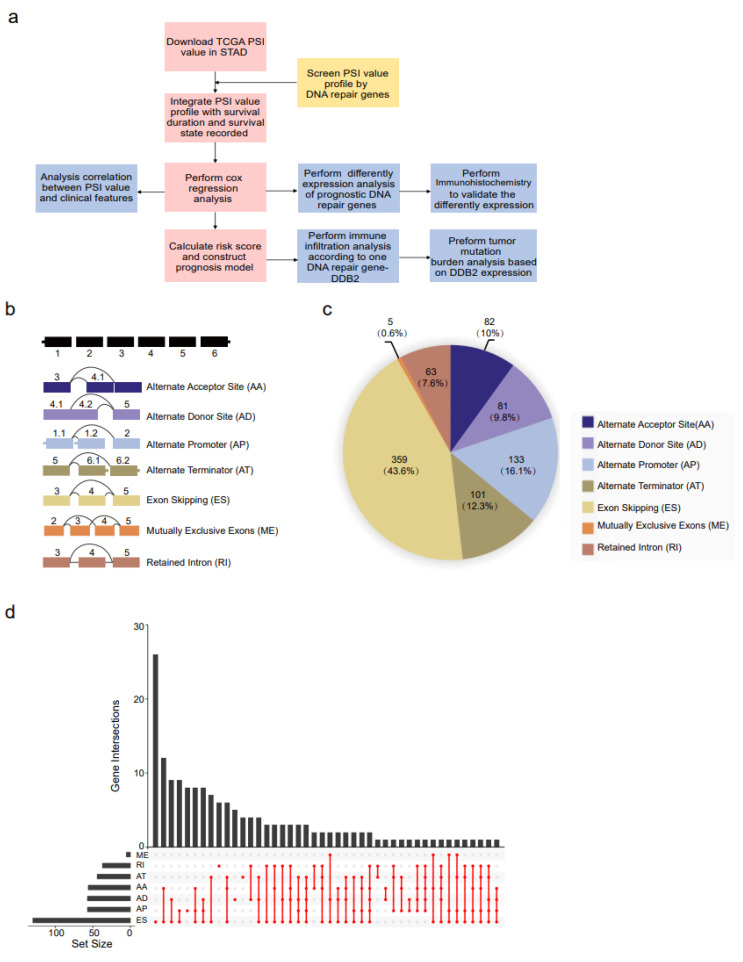
Overview of the dataset. (**a**) Flow chart of the research. (**b**) Illustration of the seven kinds of alternative splicing, namely AA, alternate acceptor site; AD, alternate donor; AP, alternate promoter; AT, alternate terminator; ES, exon skipping; ME, mutually exclusive exons; and RI, retained intron. (**c**) Number of AS events in 375 STAD patients. (**d**) UpSet plots of the seven types of optional splicing events and their parent genes.

**Figure 2 genes-14-00039-f002:**
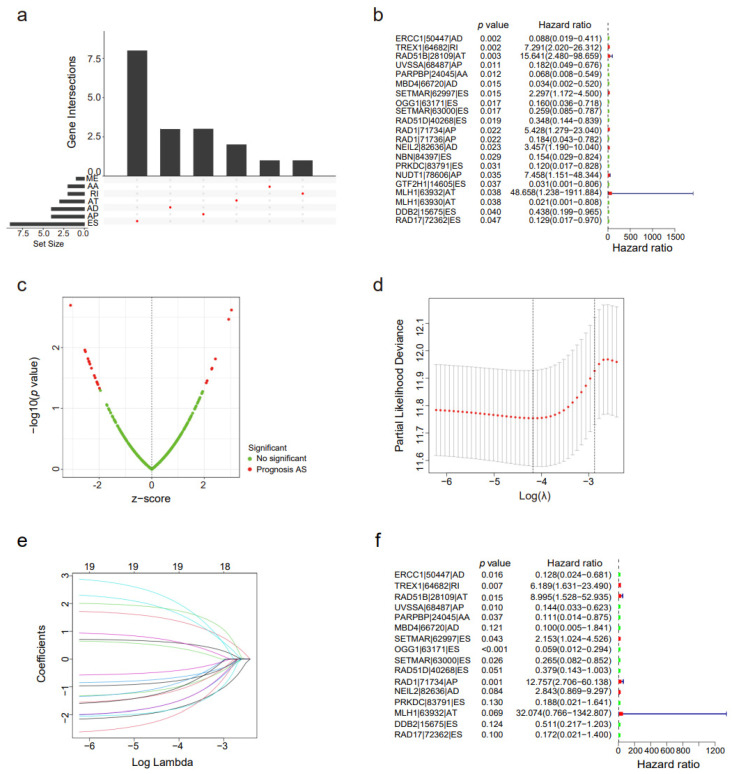
Results and constructions of the independent prognostic model. (**a**) UpSet plots of seven kinds of prognosis-related optional splicing events and their parent genes. The lower part of every plot describes the permutations of the optional splicing events; for every situation, the optional splicing event is comprised if its counterpart is full of a red dot. The higher part of the plot symbolizes the amounts of genes for every situation. Average PSI value ≤ 0.05. (**b**) The forest plot of prognosis-related AS events in the STAD cohort. (**c**) The volcano plot of prognosis-related AS events in the STAD cohort. (**d**) Lasso plot determining the amount of whole survival-related splicing events in the survival analysis by Lasso regression. (**e**) Lambda plot determining the number of whole survival-related splicing events in the survival analysis by Lasso regression. (**f**) The forest plot of the univariate Cox regression analysis for DNA repair gene alternative splicing.

**Figure 3 genes-14-00039-f003:**
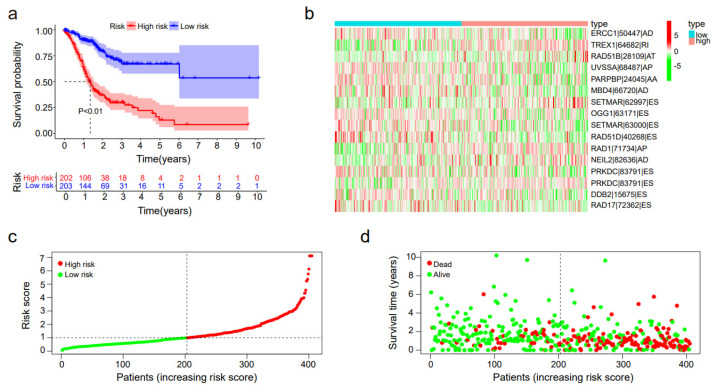
Confirmation of the risk score grounded on important prognosis-related DNA repair genes. (**a**) The Kaplan–Meier plotter of the survival analysis in which low-risk patients (purple curve) are more able to live longer than high-risk patients (red curve). (**b**) The heatmap of the expression of the final 16 selected whole survival-related splicing events from the TCGA dataset. (**c**) Scatter plot performing the risk scores of high-risk group patients and low-risk group patients. The individual inflection points of the risk score curve is shown by the dotted lines. The risk plot of both low-risk patients (green points) and high-risk patients (red points). (**d**) Scatterplot of the survival time and the risk score of both low-risk patients (green points) and high-risk patients (red points).

**Figure 4 genes-14-00039-f004:**
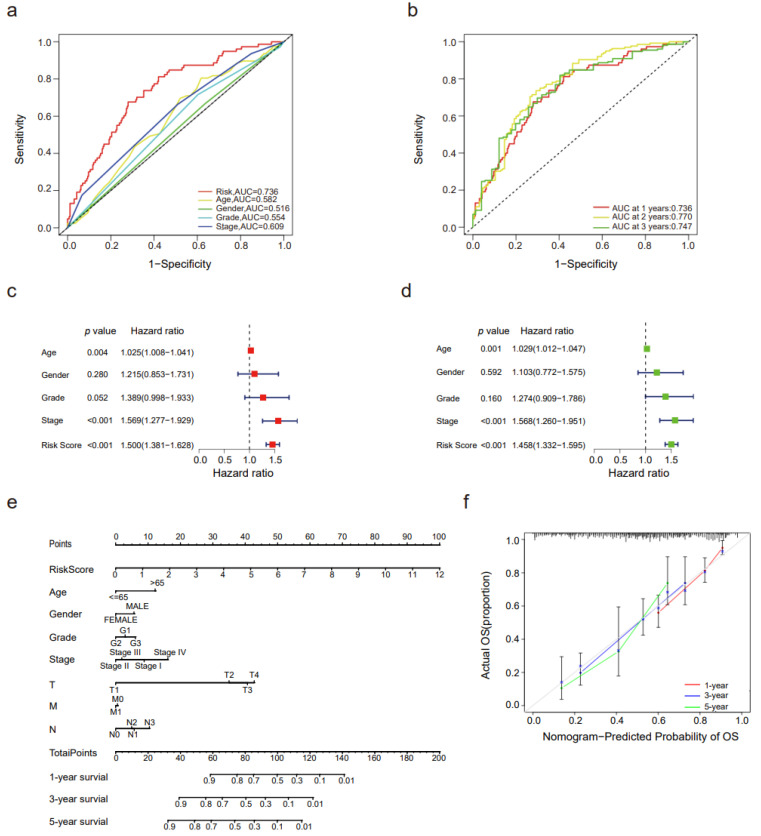
Forest plots and the nomogram for the prediction model constructed for STAD patients. (**a**) A ROC curve analysis of clinical features. (**b**) The time-dependent receiver operator characteristic curve (ROC) of survival analysis; the area under the ROC curve of the risk score was 0.736 at one year. The area under the ROC curve of the risk score was 0.736 at one year, 0.770 at two years, and 0.747 at three years. (**c**–**e**) The forest plots of univariate (**c**) and multivariate Cox (**d**) regression of the risk score and other clinical features. (**e**) The nomogram considering age, clinical stage, T stage, N stage and the risk score based on 16 prognosis-related DNA repair genes predicted the 1-year, 3-year, and 5-year survival of STAD patients. (**f**) The calibration curve for the nomogram. ROC curve analysis of the risk score at 1 year, 3 years, and 5 years.

**Figure 5 genes-14-00039-f005:**
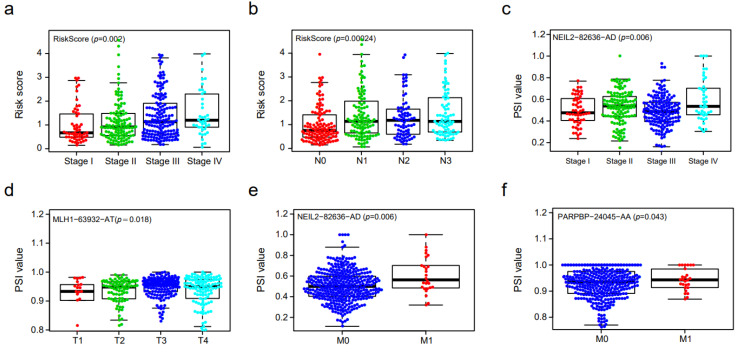
Correlation between the PSI value of prognosis-related DNA repair gene alternative splicing and clinical traits. (**a**–**f**) Box plot is shown. (**a**,**b**) Risk score in different TNM stages. (**c**,**d**) The PSI values of NEIL2-82636-AD and MLH1-63932-AT in different TNM stages. (**e**,**f**) The PSI values of NEIL2-82636-AD and PARPBP-24045-AA in different M stages.

**Figure 6 genes-14-00039-f006:**
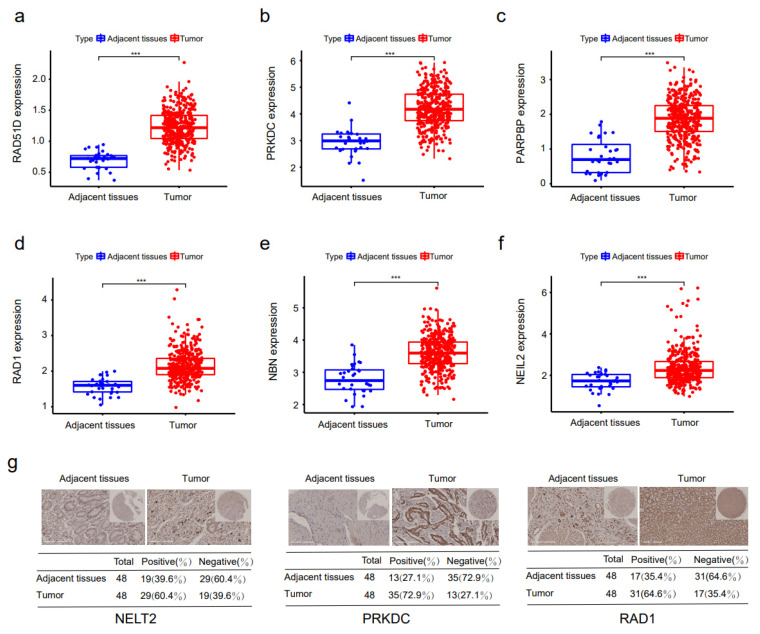
The expressing levels of DNA repair genes between STAD tumor tissues and neighboring ordinary tissues. (**a**–**f**) The mRNA expression level of host genes whose alternative splicing is obviously related to the prognosis of STAD patients’ overall survival between STAD tumor tissues and neighboring ordinary tissues. (**g**) IHC analysis of the directive genes in STAD tumor tissues and neighboring ordinary tissues. *** *p* < 0.001.

**Figure 7 genes-14-00039-f007:**
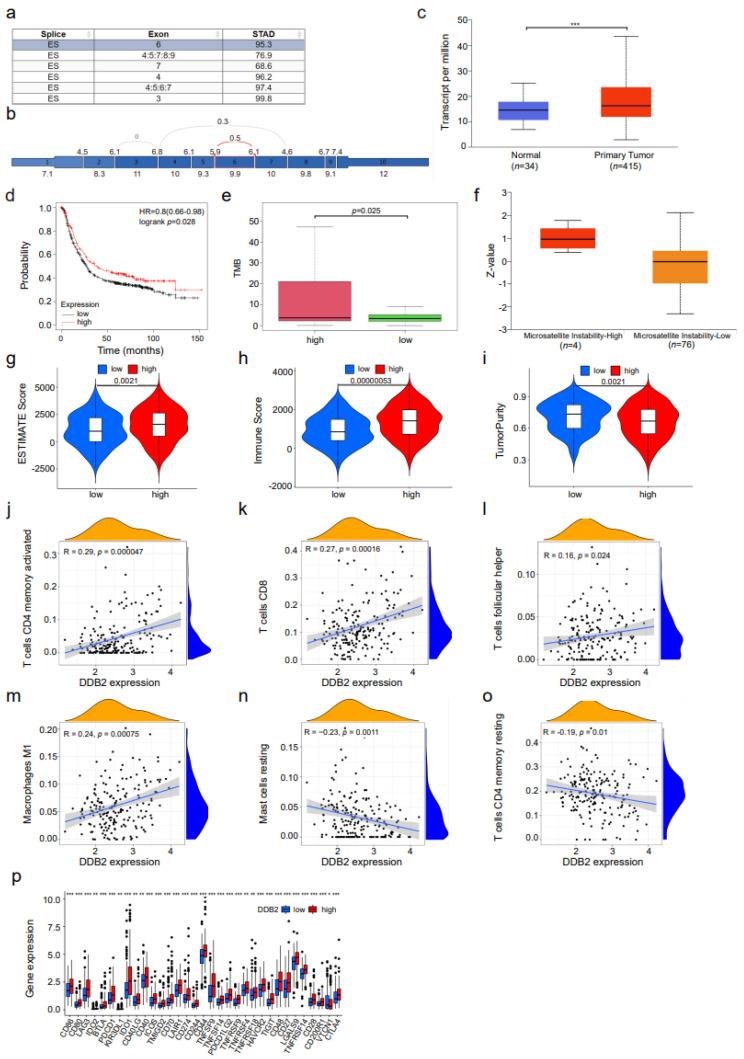
*DDB2* could be the potential prognostic biomarker in STAD patients with active immune responses and better survival. (**a**) Results table shows *DDB2* splicing type, exon, and average PSI values for STAD in the TCGA SpliceSeq. (**b**) A splice graph of the *DDB2*’s exons were shaded based on expression level and shows the selected splice event outlined in red. (**c**) The mRNA expression level of *DDB2* between STAD tumor tissues and neighboring ordinary tissues in UALCAN dataset. (**d**) Kaplan–Meier curves of OS for patients with high and low *DDB2* expression. (**e**) TMB score of *DDB2* for STAD patients between high and low *DDB2* expression. (**f**) Comparisons of MSI-H and MSI-L STAD tumors between high and low *DDB2* expression in the UALCAN dataset. (**g**–**i**) The ESTIMATES score, immune score, and tumor purity between the low *DDB2* expression group and the high *DDB2* expression group. (**j**–**o**) The correlation between the fraction of immune cells in tumor tissues and the expression level of *DDB2*. (**p**) The score of immune checkpoint genes between the high *DDB2* expression group and the low *DDB2* expression group. *** *p* < 0.001.

**Table 1 genes-14-00039-t001:** Clinical information of 443 STAD patients.

Clinical Features	Parameters	Number of Patients (%)
Gender	Male	285 (64.3%)
	Female	158 (35.6%)
Age	<60	132 (29.7%)
	≥60	306 (69%)
	unknow	5
Pathological stage	Stage I	57 (12.8%)
	Stage II	132 (29.7%)
	Stage III	183 (41.3%)
	Stage IV	44 (10%)
	unknow	27
T stage	T1 stage	23 (5.2%)
	T2 stage	93 (20.1%)
	T3 stage	198 (44.7%)
	T4 stage	119 (26.9%)
	TX stage	10 (2.2%)
N stage	N0 stage	132 (29.8%)
	N1 stage	119 (26.9%)
	N2 stage	85 (19.2%)
	N3 stage	88 (19.9%)
	NX stage	17 (3.8%)
	unknow	12
M stage	M0 stage	391 (88.3%)
	M1 stage	30 (6.8%)
	MX stage	22 (5%)

**Table 2 genes-14-00039-t002:** Differentially expressed DNA repair genes whose PSI value of their alternative splicing events is related to the overall survival of STAD patients.

Gene	Control Mean	Tumor Mean	logFC	*p* Value
PRKDC	2.935782	4.208428	1.272646	6.96 × 10^−16^
PARPBP	0.798949	1.857974	1.059025	7.81 × 10^−15^
NBN	2.78608	3.600753	0.814671	7.79 × 10^−13^
NEIL2	1.702138	2.377105	0.674968	4.81 × 10^−8^
RAD1	1.57248	2.157504	0.585024	1.09 × 10^−14^
RAD51D	0.695483	1.233019	0.537535	8.93 × 10^−19^
MBD4	2.972744	3.377801	0.405057	5.64 × 10^−6^
UVSSA	1.087983	1.445453	0.35747	2.89 × 10^−6^
RAD17	2.407257	2.654737	0.24748	0.00016
DDB2	2.259899	2.49852	0.238621	0.026581
RAD51B	0.501004	0.672451	0.171448	4.18 × 10^−7^

## Data Availability

All authors declare that all data supporting the findings of this study are available in the article. The datasets generated and analyzed during the current study are available in the TCGA, ESTIMATE algorithm, CIBERSORT, STRING, and Molecular Signatures Database repository. (https://bioinformatics.mdanderson.org/TCGASpliceSeq/, accessed on 12 May 2022); (https://string-db.org/, accessed on 22 May 2022); (https://www.gsea-msigdb.org/gsea/msigdb, accessed on 29 May 2022); (https://bioinformatics.mdanderson.org/estimate/, accessed on 2 December 2022); (http://cibersort.stanford.edu/, accessed on 2 December 2022); (https://kmplot.com, accessed on 1 December 2022); (https://Ualcan.path.uab.edu, accessed on 1 December 2022).

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
