# Peer review of "The Role of Alternative Splicing Factors, DDB2-Related Ageing and DNA Damage Repair in the Progression and Prognosis of Stomach Adenocarcinoma Patients"

_genes, 2022, doi:10.3390/genes14010039_

Round 1

Reviewer 1 Report

In this manuscript, Wang, Huang et al detect 824 alternative splicing events that originate from 166 genes and created a prognostic prediction model for STAD patients. Overall, the study is designed well and data are convincing. The following points should be addressed before the study is published.

What is the rationale for selecting the 166 genes in this study? This should be discussed in text. 

Majority of anti-cancer therapies work by causing DNA damage. Can this model predict response to first/second line chemo/radiotherapeutics if patients are stratified by risk score?  

Minor: 

Figure/text formatting is off for many figures- this should be corrected before publication. 

Author Response

Reviewer 1:

In this manuscript, Wang, Huang et al detect 824 alternative splicing events that originate from 166 genes and created a prognostic prediction model for STAD patients. Overall, the study is designed well and data are convincing. The following points should be addressed before the study is published.

Q1: What is the rationale for selecting the 166 genes in this study? This should be discussed in text.

A1: Thanks for your advices. We had added some words into manuscript to explain the rationale. We downloaded the DNA repair genes list included 219 genes totally [1]. We sorted the PSI value profile according to this gene list. At last, 166 genes were equipped with recorded alternative splicing event in PSI value profile.

Ref 1: Knijnenburg, T.A.; Wang, L.; Zimmermann, M.T.; Chambwe, N.; Gao, G.F.; Cherniack, A.D.; Fan, H.; Shen, H.; Way, G.P.; Greene, C.S.; et al. Genomic and Molecular Landscape of DNA Damage Repair Deficiency across The Cancer Genome Atlas. Cell Rep 2018, 23, 239-254.e236, doi:10.1016/j.celrep.2018.03.076.

Q2: Majority of anti-cancer therapies work by causing DNA damage. Can this model predict response to first/second line chemo/radiotherapeutics if patients are stratified by risk score? 

A2: It is a good idea. It is possible that this model could predict the response of STAD patients to chemotherapy or radiotherapy. Further research is required to provide with evidence.

Firstly, we could design Case control study. For example, one hundred STAD patients were invited into study. Then we divided them into chemotherapy sensitive group and chemotherapy resistance group according to clinical standard. Then RNA seq experiment was performed to retrieve the PSI value of prognostic DNA repair AS events. Risk score could be calculated according to the PSI value. Patients in chemotherapy sensitive group could be divided into high risk subgroup and low risk subgroup. Similiarly, Patients in chemotherapy resistance group could be divided into high risk subgroup and low risk subgroup. then we could calculate the odds ratio to reflect the correlation between risk level and chemotherapy response. Alternatively, we could perform logistic regression analysis to calculate the odds ratio and its confidence interval.

Secondly, we could design cohort study. For example, one hundred STAD patients were invited into study. Then we perform RNA seq on their STAD samples. We could calculate the risk score of STAD patients according to the PSI value of prognostic DNA repair gene alternative splicing events. We divide them into high risk group and low risk group according to risk scores. Chemotherapy treatments were performed on them to divide them into Chemotherapy resistance subgroup and chemotherapy sensitivity subgroup. Meanwhile, we could record the duration before the chemotherapy resistance appear if some patients exchange form chemotherapy sensitivity into chemotherapy resistance. Then we could calculate the risk ratio to evaluate the predict value of risk level to chemotherapy resistance. Alternatively, we could perform cox regression to obtain risk ratio and its confidence interval evaluating the correlation between risk score and chemotherapy resistance.

Case control study and cohort study deserve to perform in future to figure out the prediction value of PSI value of DNA repair genes’ alternative splicing events to chemoresistance or radio- resistance.

Minor:

Q3: Figure/text formatting is off for many figures- this should be corrected before publication.

A3: Thank for your advices. We reformed the manuscript. Please check it again.

Reviewer 2 Report

In this manuscript, the authors aim to investigate the potential roles of the alternative splicing(AS) of DNA repair genes play in the stomach adenocarcinoma with published databases. They analyzed the AS of 166 genes and constructed the prediction model with COX regression analysis. The model suggests that the AS in DNA repair genes may be an independent biomarker in the clinic.

My concerns: 1, Whether the data from different databases can be integrated and construct the prediction model.

2, Fig 2b and 2f: the data show that some AS events in DNA repair genes mean protective factor (HR<1) and AS events in 4 DNA repair genes mean risk factors(HR>1), how to explain it.

3, Fig 3a. it is unclear which types of PSI value means high risk or low risk.

4, Fig 7. whether it is necessary to integrate the data of immune infiltration with AS events in DNA repair genes. these data have nothing to do with AS of genes.

Author Response

Reviewer 2:

In this manuscript, the authors aim to investigate the potential roles of the alternative splicing(AS) of DNA repair genes play in the stomach adenocarcinoma with published databases. They analyzed the AS of 166 genes and constructed the prediction model with COX regression analysis. The model suggests that the AS in DNA repair genes may be an independent biomarker in the clinic.

My concerns: Q1, Whether the data from different databases can be integrated and construct the prediction model.

A1: We integrated the survival data and alternative splicing event to perform cox regression analysis. All of them origin from TCGA project. We did not integrate different databases to construct prediction model. In fact, this question depends on concrete condition you are facing. It triggers dispute in some cases. For example, gene chip data sets are often considered to be allowed to integrate for analysis. Whereas, the experiment condition such as temperature may produce great influence on the probe in chips. Batch effect may trigger great error when the gene chip data sets were integrated although there are some methods to cover batch effect such as sva packages in R software. As for RNA seq data set, more problems emerges when you integrate different databases for analysis due to the complex workflows to produce gene expression matrix based on original disembarkation data. Although many articles which claimed they integrate different databases have been published already. In my opinion, it should still be prudent to do it.

Q2, Fig 2b and 2f: the data show that some AS events in DNA repair genes mean protective factor (HR<1) and AS events in 4 DNA repair genes mean risk factors (HR>1), how to explain it.

A2: Cox regression analysis is applied to explore the influence that risk factors produce on risk function. Cox regression model could be constructed by survival data of patients and the level of risk factors. Survival data including survival duration and survival status. Risk factors could be continuing data or diffuse data. Risk function reflect the dead rate of people involved in cohort research. If the factors equipped with positive coefficient, the higher level of it means the higher risk facing death. Meanwhile HR>1. If the factors equipped with negative coefficient, the higher level of it means the lower risk facing death. Meanwhile HR<1. Fig 2b is univariate cox regression analysis, it provided with prognostic significant alternative splicing events. Fig 2f is multivariate cox regression analysis, alternative splicing events involved in it were optimized by lasso regression analysis to avoid overfit. Some alternative splicing events were removed. At last, 16 alternative splicing events were preserved at last multivariate cox regression model. 10 AS events were considered as protective factors and 6 AS events were considered as risk factors.

Q3, Fig 3a. it is unclear which types of PSI value means high risk or low risk.

A3: The risk score could be calculated according to the PSI value of prognostic DNA repair genes alternative splicing presented in Fig 2F. The computational formula is provided in method chapter. We calculate the risk score of each STAD patients by PSI value of 16 types of alternative splicing events. Because they were existed in multivariate cox regression models which proved that they were prognostic significantly. Higher risk score means faster rate of death according to the result of cox regression analysis. It is not advised to stratified patients into high risk group or low risk group according to the PSI value of only one alternative splicing event. In my opinion, the combination of 16 types of alternative splicing events has a better prediction ability than 1 type for prognosis.

Q4, Fig 7. whether it is necessary to integrate the data of immune infiltration with AS events in DNA repair genes. these data have nothing to do with AS of genes.

A4: More and more studies have shown that immune cells that infiltrate tumors play a crucial role in the progression and invasion of cancer. It is increasingly recognized that molecular markers of tumor infiltrating immune cells are good biomarkers for prognosis prediction. Tumour Immunotherapy has revolutionized cancer treatment and substantially improved patient outcome with regard to multiple tumour. DNA damage response (DDR) deficiency has recently emerged as an important determinant of tumour immunogenicity. A growing body of evidence now supports the concept that DDR- targeted therapies can increase the anti-tumour immune response by promoting antigenicity through increased mutability and genomic instability. Therefore, we integrate the data of immune infiltration with AS factor in DNA repair genes.

Reviewer 3 Report

The Ms entitled “Ageing and DNA repair-related alternative splicing predicts the prognosis of stomach adenocarcinoma patients” is a work that shows a risk analysis determined from the TCGA considering both Stomach Adenocarcinoma and Alternative Splicing.

Minor concerns:

·         In the abstract section the authors should consider adding more specific results of the analyzes carried out that reflect what is mentioned in the title of the manuscript.

·         The introduction of the manuscript begins by supplying background information on alternative splicing (AS), then aging and its relationship to DNA damage response and senescence induction are introduced in a logical and contextualized manner. However, when they abruptly address the epidemiological background of gastric adenocarcinoma (STAD) and the genomic analysis performed by TCGA, it forces the inclusion of the role of repair gene polymorphisms. Giving the impression that once the results were obtained, the relevance of said polymorphisms was evidenced.

·         The transcriptome sequencing data of stomach adenocarcinoma specimens recovered from patients suffering from stomach cancer with their clinical characteristics were obtained from the TCGA database, however, the authors do not indicate which filters were applied to obtain the database analyzed.

·         The authors must be clear and indicate the origin of the samples to which the IHC staining was performed and how much the TCGA analysis represents.

·         The authors must include the logical approach of each of the applied analyzes, since it is confusing to understand the origin of the databases used in the different platforms.

The discussion section is a collection of data with no relevance to adenocarcinoma of the stomach.

Major concerns:

The greatest concern regarding the manuscript is the lack of a justification that reflects the novelty of the finding to which the authors refer. It is difficult to find novelty in the results based on analysis of patient data (without knowing if they are men, women, a specific age cohort, ethnic group, stage of STAD progression, etc.) that involve alterations in repair genes and DDR, since it is based on samples of a cancer that is recognized by the authors to be diagnosed late. This late stage already involves a high degree of genomic instability that is triggered both by alterations in repair genes and by AS.

Author Response

Reviewer 3:

The Ms entitled “Ageing and DNA repair-related alternative splicing predicts the prognosis of stomach adenocarcinoma patients” is a work that shows a risk analysis determined from the TCGA considering both Stomach Adenocarcinoma and Alternative Splicing.

Minor concerns:

Q1: In the abstract section the authors should consider adding more specific results of the analyzes carried out that reflect what is mentioned in the title of the manuscript.

A1: Thanks for your advice. The specific results mentioned in the title has been added in abstract of manuscript.

Q2: The introduction of the manuscript begins by supplying background information on alternative splicing (AS), then aging and its relationship to DNA damage response and senescence induction are introduced in a logical and contextualized manner. However, when they abruptly address the epidemiological background of gastric adenocarcinoma (STAD) and the genomic analysis performed by TCGA, it forces the inclusion of the role of repair gene polymorphisms. Giving the impression that once the results were obtained, the relevance of said polymorphisms was evidenced.

A2: Thanks for your advice. We have rewritten the introduction of the manuscript.

Q3: The transcriptome sequencing data of stomach adenocarcinoma specimens recovered from patients suffering from stomach cancer with their clinical characteristics were obtained from the TCGA database, however, the authors do not indicate which filters were applied to obtain the database analyzed.

A3: The FPKM corrected RNA seq data of all the STAD samples in TCGA were downloaded to perform analysis. transcriptome sequencing data of all the STAD patients in TCGA database was downloaded without filters.

Q4: The authors must be clear and indicate the origin of the samples to which the IHC staining was performed and how much the TCGA analysis represents.

A4: Thanks for your advices. We downloaded clinical information from TCGA database included 443 STAD patients’ clinical features, which were listed in Table 1. IHC samples were purchased from SHANGHAI OUTDO BIOTECH CO., LTD, which included 30 STAD tissue samples. This study was approved by Zhejiang Taizhou Hospital, following the Helsinki Declaration. We have added the information in manuscript.

Q5: The authors must include the logical approach of each of the applied analyzes, since it is confusing to understand the origin of the databases used in the different platforms.

A5: Thanks for your advices. In the present study, we integrated the survival data and alternative splicing event to perform cox regression analysis. All of them origin from TCGA project. Specifically, we focused on DNA repair genes' AS in the TCGA STAD cohort. We screen out prognostic AS events of DNA repair genes and studied the relationships between these events and clinical features in GC patients. The TCGA SpliceSeq database was applied to download the PSI value profile of alternative splicing event. We downloaded FPKM corrected RNA seq profile from TCGA database for gene expression level analysis. Clinical data of STAD patients was downloaded from TCGA database. We retrieved the prognostic DNA repair genes alternative splicing events by cox regression. Then the risk score was calculated according to the PSI value of these prognostic DNA repair gene alternative splicing events. The distribution of expression level of host genes of prognostic DNA repair genes correlated alternative splicing events between tumour specimens and para-cancer samples were analyzed. Among all the differently expressed genes, DDB2 was selected as target for prognosis and immune infiltration. We have added an analysis flowchart in Figure 1a. Please check it again.

Q6: The discussion section is a collection of data with no relevance to adenocarcinoma of the stomach.

A6: Thanks for your advice. We have deleted the data with no relevance to GC and rewritten the discussion of the manuscript.

Major concerns:

Q7: The greatest concern regarding the manuscript is the lack of a justification that reflects the novelty of the finding to which the authors refer. It is difficult to find novelty in the results based on analysis of patient data (without knowing if they are men, women, a specific age cohort, ethnic group, stage of STAD progression, etc.) that involve alterations in repair genes and DDR, since it is based on samples of a cancer that is recognized by the authors to be diagnosed late. This late stage already involves a high degree of genomic instability that is triggered both by alterations in repair genes and by AS.

A7: Thanks for your advices. DNA damage response is a key signal transduction pathway in triggering ageing and tumour progression. Abnormal alternative splicing (AS) is associated with tumours and ageing. However, the role of AS factors associated DNA damage repair and ageing in stomach adenocarcinoma (STAD) remain unclear. In the present study, we focused on AS of DNA repair genes in TCGA STAD cohort. We screened out prognostic AS events of DNA repair genes and constructed prognostic prediction model. More importantly, we found a novel AS factor DDB2 associated with ageing and DNA repair, which was related with high TMB, high MSI, active immune responses and better survival. And DDB2 maybe provide potential biomarkers for prognosis and tumour therapy in STAD.

Furthermore, we downloaded clinical information from TCGA database included 443 STAD patients’ clinical features (including TNM stage, Gender and Age). The detail data were listed in Table 1. These data covered four pathological stage (Stage I:12.8%, Stage II:29.7%, Stage III:41.3%, Stage IV:10%), five T stage (T1 Stage:5.2%, T2 Stage:20.1%, T3 Stage:44.7%, T4 Stage:26.9%, TX Stage:2.2%), five N stage (N0 Stage:29.8%, N1 Stage:26.9%, N2 Stage:19.2%, N3 Stage:19.9%, NX Stage:3.8%), three M stage (M0 Stage:88.3%, M1 Stage:6.8%, M2 Stage:5%) , Gender (male: 64.3%, female: 35.7%), and Age (<60: 29.7%, >=60:69%). Early stages and late stages STAD cancer patients’ data were included and these sample may be involved different degree of genomic instability.

Round 2

Reviewer 2 Report

The revision makes the MS better and the authors solved my concerns.

Reviewer 3 Report

I think that the authors did a great job, meeting the requirements of the review. I am satisfied with the current version of the manuscript, I think it has improved considerably.